# Moderate Constraint Facilitates Association and Force-Dependent Dissociation of HA-CD44 Complex

**DOI:** 10.3390/ijms24032243

**Published:** 2023-01-23

**Authors:** Ziyang Yao, Jianhua Wu, Ying Fang

**Affiliations:** Institute of Biomechanics, School of Biology and Biological Engineering, South China University of Technology, Guangzhou 510006, China

**Keywords:** CD44, HA, MD simulation, structure–function relation, mechanical regulation on receptor-ligand interaction

## Abstract

Binding of cell surface glycoprotein CD44 to hyaluronic acid (HA) is a key event for mediating cell adhesion, motility, metastasis, inflammatory responses and tumor development, but the regulation mechanism and its molecular basis under diverse mechanical constraints remain unclear. We herein investigated interaction of CD44 HABD (HA binding site domain) to HA through free and steered molecular dynamics (MD) simulations as well as atomic force microscope (AFM) measurement using different constraints on HA. The middle, two ends or both of the constrained HA chains were fixed for MD simulations, while one and two biotin–avidin linkage or physical absorption were used to immobilize HA on substrates for AFM experiments, to model HA chains with low, moderate and high HA flexibilities, respectively. We found that binding of CD44 to moderate fixed HA was possessed of a better thermo-stability, a lower mechanical strength and a higher dissociation probability, while higher adhesive frequency, smaller rupture force and shorter lifetime were assigned to CD44 on the two biotin-immobilized HA rather than one biotin-immobilized or physically absorbed HA on substrates, suggesting a moderate HA flexibility requirement in favor of association and force-induced dissociation of CD44-HA complex. Tensile-induced convex conformation of HA chain was responsible for reduction of complex mechano-stability and did inversely a shrunken CD44 HABD under stretching; transition from catch bond to slip bond governed CD44-HA interaction. This study uncovered the regulation mechanism and its molecular basis for CD44-HA affinity under diverse mechano-microenvironments and provided a new insight into CD44-HA interaction-mediated cell inflammatory responses and tumor development.

## 1. Introduction

Interaction of cell surface glycoprotein CD44 with hyaluronic acid (HA) mediates cell adhesion, motility, metastasis, inflammatory responses and tumor development [1,2]. CD44 bound with HA connects the extracellular matrix (ECM) and the cellular cytoskeleton and then enable cells to sense and transmit mechano-signals from the matrix [3], while organizing receptor platforms on the cell surface to activate multiple downstream signaling pathways [4]. Some regulators for binding of CD44 with HA are known as the N-glycosylation of CD44 [5], the presentation form and molecular weight (MW) size of HA [6], the mechanical microenvironment and so on [7,8], but less knowledge exists especially in the mechanical regulation mechanism and its molecular basis for CD44 binding affinity with HA.

CD44, a well-known cancer stem cell (CSC) biomarker [9], is expressed on leukocyte [10] and tumor cells [11] and involved crucially in inflammatory leukocyte homing, tumorigenesis and cancer progression [12,13]. CD44 exists in the standard form (CD44s) or multiple variant isoforms (CD44v), but each form consists of four major regions, a ligand binding domain, a highly glycosylation proximal membrane region (stalk region), possibly including the inserted variant exon products, a transmembrane domain and a cytoplasmic tail (CT); the recognizable ligands of CD44 include HA, E-selectin, osteopontin (OPN), collagen, fibronectin and chondroitin, and the ligand binding domain of CD44 is called HA binding domain (HABD). Because of that, HA serves as the most crucial CD44 ligand among those CD44 ligands mentioned above and is highly conserved in both CD44s and CD44v [14]. CD44 bound with its ligands may undergo a conformational change in favor of binding of cytoskeletal linker protein ankyrin, ezrin, radixin and moesin (ERM) to CD44 C-terminal tail for downstream signaling first [15,16] and then a sequential proteolytic cleavage in either the ectodomain and intracellular domain to regulate cell motility, gene expression and proliferation [17,18]. HA is composed of repeating disaccharide units consisting of β(1-4)-D-glucuronic acid (GlcUA) and β(1-3)-N-acetylglucosamine (GlcNAc) but is an extraordinarily versatile polymer with a wide range of molecular weights from 6 kDa to more than 20 MDa [19]. Diverse HA polymers have different roles in physiological and pathological processes of leukocytes and tumor cells [20]. Generally, HA with high molecular weight exhibits anti-inflammatory and anti-angiogenic functions, while the HA of a low molecular weight possesses pro-inflammatory, pro-angiogenic and tumorigenic properties [21,22].

The structure of CD44 HABD from either X-ray crystallography and NMR spectroscopy shows that four additional β-strands from the flanking N- and C-terminal extensions are linked by the third disulfide bridge and form a lobular elongation to the lectin-like (Link) module and contributed to binding with HA [23], unlike most HA binding proteins that interact with ligand only via a Link module consisting of two antiparallel β sheets and two α helices and being stabilized just by two conserved disulfide bridges [24]. The crystal structure of mouse CD44 with HA exhibits that the multiple loops on HABD are in contact with HA to form a stable binding groove dominated by interfacial hydrogen bonding, and the 13 involved residues (Arg^45^, Tyr^46^, Cys^81^, Arg^82^, Tyr^83^, Ile^92^, Asn^98^, Ile^100^, Cys^101^, Ala^102^, Ala^103^, His^105^ and Tyr^109^) on Link module and Arg^155^ on C-terminal extension are identified as critical (Figure 1) [25]. HA binding, mutation and shear force can induce conformational change of CD44 from the ordered (O) conformation to the partially disordered (PD) one, along with enhancement of HA affinity [26,27].

It is believed that the stiffness of the HA-coated substrate correlates with the HA weight and substrate surface roughness as well as the HA-immobilized manner and serves as a regulator for CD44 binding to HA [28,29,30]. Matrix sclerosis, a common complication in cancer progression [31], will affect the flexibility of the concomitant HA significantly, causing a change of biological function of HA-enriched ECM and further promoting tumor cell metastasis [32]. The ECM stiffness is demonstrated as a pivotal factor in interaction of CD44 with HA [18,33]. In vivo HA is embedded in the ECM, possibly causing reduction of both flexibility and movement freedom of HA because of the ECM constraint on HA [22]. In vitro experiments have shown that a stiffer HA hydrogel may facilitate the adhesion and migration of glioblastoma multiforme [18], but too hard or soft HA matrix is not in favor of CD44-mediated adhesion and movement of human gastric cancer cells [30]. A moderate matrix stiffness is also required for an optimal Rho-associated protein kinase (ROCK)-mediated mobility of breast cancer cells also [34]. The influence of molecular weight on HA affinity still remains unclear. From surface plasmon resonance (SPR) analysis for HA-coated nanoparticles, Mizrahy et al. found that HA affinity increased significantly with the weight in a range from 31 to 700 kDa [6]. Meanwhile, Kim et al. stated that HA binding to CD44s is indiscriminate in incubating either 25 kDa or 700 kDa HA with breast cancer cells [35]. Different mechanical constraints on HA in the two works may be a rational explanation for these two inconsistent statements. However, it still remains unclear how the manner of the mechanical constraint regulates flexibility, allostery, biological function and their molecular basis of HA chains at static state or tensile force.

We herein performed atomic force microscope (AFM) experiments at single molecular level, free and steered molecular dynamics (MD) simulations with the structure of CD44 HABD-HA complex to investigate the mechanical regulation mechanism and molecular basis for CD44 interaction under various mechanical microenvironments. The middle, two ends or both of the constrained HA chains were fixed for MD simulations, while one and two biotin–avidin linkages or physical absorption were used to immobilize HA on substrates for AFM experiments, to model HA chains with low, moderate and high HA flexibilities and freedom, respectively. The present data from AFM measurements and MD simulation supported each other, and a constraint (on HA)-dependent CD44 binding to HA along with an allostery of the complex might be helpful to understanding effects of mechanical constraint on CD44-HA interaction and its downstream cellular signaling, inflammatory responses and tumor development under a mechano-microenvironment. Additionally, the novel MD-based computer strategy presented in this work might find its application in other different molecular systems under the mechano-microenvironment.

## 2. Results

### 2.1. Mechanical Constraint Might Upregulate Structural Stability of CD44-HA Complex

To gain a rational conformation of CD44-HA complex in a physiological environment, 40 ns equilibrium was performed thrice for the complex, which was regarded as the equilibrated one; because of that, the time course of root-mean-square deviation (RMSD) of Cα-atoms was fluctuated on its respective stable plateaus with small relative derivation for each run (Appendix A). We found that binding of CD44 to HA was dominated by the hydrogen bonds (H-bonds) with average occupancies (>40%) in the top nine, and these H-bonds were contributed by GlcUA1180 with Arg^155^, GlcUA1178 with Arg^82^ and Cys^81^ as well as Arg^45^, GlcNAc1177 with Tyr^46^ and Ile^100^, GlcUA1176 with Arg^45^, and GlcNAc1175 with Asn^115^ and Tyr^119^ (Appendix A in Appendix A). According to the frequency distribution of H-bond number in thrice equilibrium simulation (Appendix A), the best stable equilibrated structure from Run 3 was selected as the initial conformation for subsequent free molecule dynamics (FMD) simulations and steered molecular dynamics (SMD) simulations.

Diverse attachments of HA to ECM in a physiological environment would lead to different mechanical constraints on HA. We herein used four molecular systems, named S0, S1, S2 and S3, which imitated HA chains either in solution or loosely, moderately and tightly attached to ECM, respectively, to examine effects of mechanical constraints on HA affinity to CD44 (Materials and Methods). Of these systems, S0 was the same as the equilibrated system with no mechanical constraint on complex, S1 was set up just through fixing the C2 atom of HA GlcUA1178 of the equilibrated complex, and S2 had fixed two C2 atoms, one on HA GlcNAc1175 and the other on HA GlcNAc1181, while S3 was built up through fixing three C2 atoms on GlcNAc1175, GlcUA1178 and GlcNAc1181 of CD44-ligated HA, respectively. FMD simulation was performed on each system thrice over 100 ns with time step of 2 fs.

From the time course of the Cα-RMSD of CD44-HA (Figure 2), we obtained that the structure of CD44-HA complex in each system had a good thermal stability because the Cα-RMSD of the complex was located at a plateau with a slight fluctuation (Figure 2a). For CD44-HA complex in systems S0 and S1, their Cα-RMSD was located at a higher level in comparison with those in other two systems, S2 and S3, and the mechanical constraint-induce down-movement of the Cα-RMSD plateau of CD44-HA occurred visibly in systems S2 and S3 but not in system S1 (Figure 2a) and came from HA rather than CD44 (Figure 2b,c), suggesting a mechanical constraint (in an appropriate manner)-enhanced structural stability of CD44-HA in system S2 or S3. Additionally, a mechanical constraint-reduced flexibility of the bound HA in system S2 or S3 might cause a change of HA affinity to CD44.

### 2.2. Moderate Mechanical Constraint on HA Facilitates Binding of HA to CD44 via a HA Flexibility-Dependent Allostery of CD44 and an Enhancement of Interfacial H-Bond Interaction

Besides the structural stability and flexibility, the affinity of the bound HA to CD44 might be regulated by the manner of the mechanical constraint on HA. We counted H-bonding events on binding interfaces and read the interaction energy of the complex one by one during each FMD run of 100 ns for each of system S0, S1, S2 and S3. Variations of the mean H-bond number (***N***_HB_), the mean interaction energy (***E***), and the dissociation possibility (***P***_D_) of complex for three runs over 100 ns versus the mechanical constraint (Figure 3a–c) showed that mechanical constraint caused an increase in ***N***_HB_ but a decrease in either ***E*** or ***P***_D_, demonstrating a constraint-induced enhancement of interaction between CD44 with the fixed HA, in comparison with the case of no fixed C2 atom in HA. While increasing the number of the fixed atoms on HA, ***P***_D_ decreased first and then increased (Figure 3c), and so did ***E*** (Figure 3b), while ***N***_HB_ increased first and then decreased (Figure 3a). The threshold point was shared by ***P***_D_, ***N***_HB_, and ***E*** in system S2, as it should be. These results were consistent with each other; because of that, the more the H-bonds there are across the binding site, the lower the interaction energy between CD44 and HA and the less the dissociation possibility of the complex. This biphasic mechano-constraint-dependent dissociation of CD44 from HA meant a transition from constraint-enhanced to constraint-weakened CD44-HA interaction and suggested that a moderate mechano-constraint in an appropriate manner such as that in system S2 might facilitate binding of CD44 to the bound HA (Figure 3c).

To uncover the dynamics mechanism for the mechanical regulation on CD44-HA interaction through fixing one or more C2 atoms on HA, we analyzed the C-RMSF per HA GlcUA or GlcNAc saccharide and the occupancies of interfacial H-bonds from three FMD runs of 100 ns for each of the systems S0, S1, S2 and S3. It was found that HA flexibility increases when fixing the C2 atom on GlcUA1178 alone but decreases if the two C2 atoms on GlcNAc1175 and GlcNAc1181, together with or without the C2 atom on GlcUA1178, were fixed (Figure 3d), while these different constraints on HA had less effects on the flexibility of bound CD44 (Appendix A). Additionally, three H-bonds, one from GlcUA1180 with its partner Arg^155^ and others from GlcUA1178 with its two partners Arg^82^ and Arg^45^, had higher survival rates and should be deeply involved in thermal stability of the complex with or without constraints on HA (Figure 3e); the H-bond between GlcUA1180-Arg^82^ was strengthened just in S1 system and then contributed to complex stability (Column 1 and 2 in Figure 3e), while the other three bonds from GlcUA1178, with its partner Arg^81^, and GlcNAc1177, with its other two partners Tyr^46^ and Ile^100^, were responsible for the thermal stability of the complex in S2 and S3 systems (Column 3 and 4 in Figure 3e). It suggested that this mechanical constraint-regulated H-bonding interaction on the binding site was relevant to the constraint-induced change of HA flexibility, and a moderately mechanical constraint such as that in system S2 would make HA affinity to CD44 high through enhancing interfacial H-bonding. The reason may be that fixing the two C2 atoms at the two ends of HA might make HA gain a moderate flexibility, which was required for forming an HA conformation in more favor with CD44. An overlarge flexibility of HA did not facilitate formation of a stable complex, while a stiff HA could not deform sufficiently to match with CD44 better under thermal excitation, leading to a biphasic mechano-constraint-dependent HA affinity to CD44.

The constraint-induced change in HA affinity to CD44 (Figure 3c) should be a reflection of thermo-excitation-driven conformation evolution of the complex. We observed from three FMD runs of 100 ns for each system (S0, S1, S2 and S3) that the mean cross angle ϕ at Tyr^46^ C_α_-atom (the triangle vertex in the bag-like CD44 HABD) (Figure 4a) took its maximum of about 88.5° in system S2 instead of the other three systems (Figure 4b), while the mean HA bend angle θ (Figure 4c) had a reduction of about 6° in system S2 and about 20° in system S0 or S1 in comparison with the bend angle of 174° in system S3 (Figure 4d). In consistence with the constraint-mediated change of the mean cross angle ϕ, the distance (*l*) between two C_α_-atoms of Arg^155^ and Ile^100^ took its maximum in system S2 instead of the other three systems (Figure 4e) and so did the SASA of HABD binding site (Figure 4f). This suggested that under mechanical constraint on HA with two fixed ends in system S2, the chain-like HA bound with CD44 would bend slightly to form a conformation in favor with CD44 while the bag-like CD44 HABD became more open to greet squeezing of the HA into the HABD bag and did inversely both HABD and HA in the other three systems.

### 2.3. Manner of Constraint on HA Regulates Mechanical Strength and Stability of CD44-HA through Pull-Induced Allostery of the Complex

A befitting mechano-constraint on HA might enhance the mechanical strength of CD44-HA complex. To examine it, we performed the force-ramp SMD simulation thrice at pulling velocity of 5 Å/ns until CD44 dissociated from HA for each of the systems S1 and S2 as well as S3 (Materials and Methods). The C_α_-atom of CD44 C-terminal was chosen as the steered point to model mechano-signaling from transmembrane protein CD44 to HA-coated extracellular matrix under mechano-microenvironments [36].

The representative force–time curves showed that the tensile force on the complex increased quickly but oscillated with pull-time until it reached the force peaks at about 12.5 ns for system S1, 8 ns for system S2 and 9.5 ns for system S3, respectively (Figure 5a–c). Additionally, the interfacial area (buried SASA) of the complex was located almost at a long plateau before slumping for systems S1 and S3 (Figure 5d,f), but not for S2 (Figure 5e). Accordingly, of the detected eight important H-bonds on the binding site, the five H-bonds from GlcUA1180 with Arg^155^, GlcUA1178 paired with Cys^81^ and Arg^82^, and GlcNAc1177 paired with Tyr^46^ and Ile^100^ were responsible for preventing pull-induced complex dissociation in each system and so was the H-bond from GlcUAc1175 with Asn^115^ in both the systems S1 and S3 (Figure 5g–i). The above observation hinted that the mechanical constraint on HA regulated not only the dissociation pathway but also the mechanical strength of CD44-HA complex.

The rupture forces, the force peaks of the force–time curves (Figure 5a–c and Appendix A), were read from “force-ramp” runs thrice to be 313 ± 10, 204 ± 16 and 289 ± 11 pN with rupture times of 11.55 ± 0.90, 8.07 ± 1.68 and 10.13 ± 1.19 ns for the systems S1, S2 and S3, respectively (Figure 5j,k), while the dissociation times were 13.64 ± 1.26, 8.933 ± 1.20 and 11.47 ± 1.51 ns (Figure 5d–f,l and Appendix A). This suggested that point constraint on the middle of HA, such as system S1 and system S3, would enhance the mechanical strength of the complex and delay the pull-induced breakage of the complex in comparison with system S2. In other words, the point constraint on the middle of HA with or without fixing two ends of HA might make the complex have a higher mechanical strength of complex to resist external force-induced structural damage, in comparison with the constraint case of two fixed HA ends in system S2.

It was obtained from force-ramp MD runs that the pull-induced conformational change was dependent on the mechanical constraint on HA. Along the pull-induced dissociation pathway, the flat bound HA at the initial or equilibrium state would become significantly concave in system S1 (Figure 6a) or convex in system S2 (Figure 6b,d) when passing through a pull time of about 2 ns, but persistently retain its flat conformation in system S3 (Figure 6c,d), possibly resulting in these different pull-induced dissociation pathways of CD44-HA complex (Figure 5) (Appendix A). Additionally, like a latch for the affinity shift of CD44 [27], H-bond between Glu^52^ and Tyr^166^ formed as linker of the Link modulus and the extensive one and should be responsible for the mechanical stability of CD44 under pulling. It was observed that the H-bond of Glu^52^-Tyr^166^ would break down at pull time of about 6 ns for system S1 and system S3 when pulling force reached about 180 pN, followed by unfolding of CD44 and the β8-β9 sheets in C-terminal extensive module away from the Link module (Figure 5g,i and Figure 6a,c,e,f), but these phenomena did not happen until CD44 dissociated from HA in system S2 (Figure 5h and Figure 6b).

These observations suggested that the mechanical strength of CD44-HA was regulated by the manner of mechanical constraint on HA through a pull-induced allostery of the complex. In the case of a center point constraint manner for HA chain in the system S1 and S3, the tensile force would mainly focus on the neighborhood of HA GlcUA1178 first, then spread out to two ends of HA, and lastly cause a local stretching of CD44, meaning a synergistic resistance manner for all interfacial H-bonds in facing tensile force from CD44 C-terminal. As a result, this pull-induced unfolding of CD44 further lengthened dissociation time and enhanced the rupture force for the complex in system S1 or S3. Inversely, under pulling, the bent and convex HA in system S2 should depart from its conformation in favor with CD44, leading to a significant reduction of mechanical strength of CD44-HA. In addition, our results not only demonstrated in detail force-induced two-state transition of CD44-HA complex from O conformation with low affinity to PD conformation with high affinity [26,27], but also revealed the underlying regulatory mechanism and molecular structural basis.

### 2.4. Tensile Force Regulates HA-CD44 Interaction via Force-Induced Conformational Change in a Mechanical Constraint (on HA)-Dependent Manner

The mechanical stability of CD44-HA complex was required for cells to sense the extracellular mechano-microenvironment and trigger multiple downstream intracellular signaling [4,12]. We herein examined the mechanical stability of the complex through “force-clamp” SMD simulations of 40 ns thrice at a constant tensile force of 25 pN for systems S1, S2 and S3 (Materials and Methods). Plots of the mean interfacial H-bond number ***N***_HB_, interaction energy ***E*** and dissociation probability ***P***_D_ of the complex from three runs over 40 ns versus system S1, S2 or S3 showed that ***N***_HB_ decreased first and then increased with the number of the fixed C2 atoms on HA (Figure 7a) and did inversely ***E*** and ***P***_D_ (Figure 7b, c), exhibiting a transition from constraint-weakened to constraint-enhanced interaction of HA with CD44 under stretching. However, the tensile force enhanced significantly the interaction of HA with CD44, especially in systems S1 and S3 (Figure 7b), in comparison with the data from FMD runs (Figure 3b). So, the interfacial H-bond numbers (***N***_HB_) of the complex in systems S1, S2 and S3 were 5.55 ± 0.03, 5.81 ± 0.02 and 5.62 ± 0.04 at static state but increased to 7.61 ± 0.03, 6.45 ± 0.04 and 8.18 ± 0.04 (Figure 3a and Figure 7a), respectively, and, at tensile force of 25 pN, the dissociation possibility of CD44-HA complex took its maximum in system S2 or minimum in system S3 instead of system S1 (Figure 7c), showing that the fixed point-constraint on the chain-like HA middle rather than the ends facilitated HA-CD44 interaction through enhancing interfacial H-bonding.

In comparison with the results from FMD simulations (Figure 3e), we found from “force-clamp” SMD simulations at tensile force of 25 pN that, of all interfacial nine involved H-bonds (Figure 7f), the H-bond from GlcUA1178 with Arg^82^ and Cys^81^, and GlcNAc1177 with Ile^100^ were enhanced for each system, while the H-bonds from GlcNAc1179 with Arg^82^ were more stable in both the systems S2 and S3 but formed newly in system S1; the H-bond from GlcUA1180 with Arg^155^, GlcNAc1177 with Tyr^46^ and GlcNAc1175 with Asn^115^ were intensified significantly in both the systems S1 and S3 but with vanishing of H-bond of GlcUA1180 with Arg^82^ and slight weakening of GlcUA1178 with Arg^45^ in system S1 (Figure 3e and Figure 7f). These data provided a statement for tensile force-enhanced binding of HA to CD44 via toning up interfacial H-bonding in an HA middle-fixed constraint-dependent manner, because most of the force-enhanced H-bonding events occurred in systems S1 and S3.

Different from the flexibility-dependent interfacial H-bonding of the complex under thermal excitation (Figure 3), tensile force-induced allostery of the complex should be responsible for the force-enhanced binding of HA to CD44 in a constraint (on HA)-dependent manner. From “force-clamp” simulations of 40 ns thrice at tensile force of 25 pN, the mean cross angle ϕ at Tyr^46^ C_α_-atom (Figure 7d) was read to be 79.15°, 85.28° and 81.33°, which became smaller than 86.35, 88.5 and 87.31 from FMD simulations for systems S1, S2 and S3 respectively, demonstrating a conformational change of CD44 under stretching; the mean HA bend angle θ (Figure 7e) took values of 203.5°, 165° and 174.5° for systems S1, S2 and S3 (Figure 7e), showing a middle-fixed constraint-induced concave HA chain and an end-fixed constraint-induced convex one under stretching. A moderately narrow binding pocket of CD44 might have high affinity to HA under pulling, but an over-convex HA might not endear itself to CD44. As a result, the force-induced enhancement of interfacial H-bonding mainly occurred in systems S1 and S3 with HA middle-fixed constraint but did not in system S2 with HA end-fixed constraints.

### 2.5. Mechanical Regulation on Adhesion Frequency, Rupture Force and Lifetime of CD44 on HA-Coated Substrates with and without Streptavidin-Treatment

To obtain experimental support for the above-mentioned results from MD simulation, we here measured adhesion frequencies, rupture forces and lifetimes of CD44 on three different 5 kDa HA-coated substrates through AFM experiments (Materials and Methods). These substrates were the functionalized Petri dishes, on which 100 μg/mL HA was anchored by one and two biotin–avidin covalent linkages or physical adsorption and used to model the one-, two- and three-point (on HA) constraints in MD simulations herein (Figure 8a). Through AFM experiments, the elastic modulus of three different HA-coated substrates were measured to be 14,396 and 17,468 or 24,597 (Pa), respectively (Figure 8b). These elastic modulus data said that, on the three different substrates, the most mechanical constraint on HA come from physical absorption instead of biotin–avidin linkage, leading to flexibility reduction or rigidity enhancement of the physically absorbed HA.

Our AFM measurements exhibited that the frequencies of CD44 adhesion to HA-coated substrates with one and two biotin–avidin linkers or nothing were 13.6% and 19.2% or 14.2%, respectively (Figure 8c), suggesting that HA-coated substrate with moderate rigidity (or flexibility) instead of low or high rigidity was in favor with CD44 (Figure 8b,c). These data were in good consistence with our results from FMD simulations (Figure 3a–c). Additionally, the rupture forces of 22, 18 and 20 pN about were detected from AFM experiments for CD44 bound to HA on substrates with one and two biotin–avidin linkers or nothing, respectively, meaning a low mechanical strength of CD44-HA complex on the flexibility (or rigidity)-moderate substrate rather than flexibility-high or low one (Figure 8d). This regulation of HA-coated substrate rigidity on the mechanical strength of CD44-HA would support the prediction results of the complex rupture force from “force-ramp” SMD simulations mentioned above (Figure 5) also, because the mechanical constraints for HA coated on the substrates was similar to those for HA in the molecular dynamic systems S1 and S2 or S3.

The plots of the lifetime of a single bond between CD44 and HA against tensile force (Figure 8e) demonstrated that, under each of the three different HA-immobilized conditions, the adhesive molecule bond lifetime increased first and then decreased with tensile force, exhibiting a transition from catch-bond to slip-bond with a tensile force threshold point of about 20 pN. The catch bond mechanism of CD44-mediated cell rolling was demonstrated in previous work [27,37], and the biphasic force-dependent CD44 dissociation from HA had been observed in many different adhesive molecule systems, such as selectins with PSGL-1 [38] and β2-integrin with ICAM-1 [39], as well as GBIbα with vWF [40]. Additionally, the AFM measurement showed that the lifetime of the complex on the rigidity-moderate substrate with two biotin–avidin linkages was significantly shorter than those on the other two types of substrates with one biotin–avidin linkage or nothing for each tensile force, possibly exhibiting a support to the “force-clamp” SMD simulation results at tensile force of 25 pN (Figure 7).

## 3. Discussion and Conclusions

CD44-HA interaction-mediated adhesion was the first step in the cascade of either inflammatory response of leukocytes or metastasis of cancer cells [12,13]. Together with blood shear stress, matrix stiffness and molecular weight, as well as immobilized manner of HA, were known as the dominant regulators for CD44 binding to HA [18,28,30], but the involved regulation mechanism and its molecular structure basis remain unclear so far. In the present work, we demonstrated the biological effect of HA-immobilized constraint, which regulated CD44 affinity with HA through changing flexibility or stiffness of the HA matrix. A loose binding site center was required for HA to be in favor with CD44 at static state.

Previous work for adherence of gastric cancer cell adhesion to HA matrix with different Young’s modulus argued that a stiffness-reduced adhesion event would ensue from a stiffness-enhanced one [30]. This biphasic regulation of HA matrix stiffness was supported by the present data from either AFM single molecular measurement or FMD simulation. Complex of CD44 with the end-fixed HA instead of the middle-fixed ones would possess stronger interfacial H-bonding and a lower dissociation probability, while higher adhesive frequency was assigned to CD44 on the two biotin-immobilized HA rather than one biotin-immobilized or physically absorbed HA on substrates. The data from both MD simulations and single molecular AFM measurements supported each other and argued the requirements of an extension conformation and a moderate flexibility or rigidity for the end-fixed but center-loosed HA in favor with CD44. Meanwhile, stronger adhesion ability and specific physiological function were assigned to the immobilized LMW-HA instead of the HMW-HA and soluble LMW-HA [36], meaning a moderate stiffness of the LMW-HA in a CD44-favorite immobilized manner, such as those in our present work, and suggesting a crucial immobilized manner-induced change of HA flexibility (Figure 3d and Figure 8b) in addition to the HA MW-dependent elastic modulus of the HA matrix [29,30].

Our data from single molecular AFM experiments showed that a smaller rupture force and a shorter lifetime were assigned to CD44 on the two biotin-immobilized HA rather than one biotin-immobilized or physically absorbed HA (Figure 8d,e), suggesting that a moderate stiffness HA-coated substrate might not be in favor of a better mechano-stability. However, transition from catch-bond to slip-bond was responsible for interaction of CD44 with HA in each HA-immobilized manner (Figure 8e). The above statements were well argued by our SMD simulations, from which the complex of CD44 with the end-fixed but center-loosed HA rather than the two center-fixed ones was obtained to have a weaker mechano-stability for its lower mechanical strength and a higher dissociation probability (Figure 5j and Figure 7c). The reason might be that the tensile force-induced convex conformation of HA (Figure 6b and Figure 7e) was responsible for the mechanical stabilization damage of the complex of CD44 with the end-fixed but center-loosed HA; because of that, an over-bent or stretched HA chain might not endear itself to CD44. Under pulling, unfolding of CD44 or extension of the C-terminal loop ensued from breakage of the H-bond between Tyr^166^ and Glu^52^ for CD44 bound with the center-fixed HA rather than the end-fixed but center-loosed HA, leading to not only an increase in rupture force but also different force-induced dissociation pathways in a constraint (on HA)-dependent manner (Figure 5). Additionally, previous results from AFM measurement stated that the rupture force of recombinant CD44-HA bond increased first and then decreased with the increase in HA molecular weight (5−2192 kDa) [28], while the molecular weight was closely related to the elastic modulus of the HA matrix [29,30]. This statement above was not in consistence with the present result, and the reason might lie not so much in the MW-dependent elastic modules of the HA matrix as in the HA immobilized manner.

Involved in the constraint-induced change of HA infinity with CD44, interfacial H-bonding events of the complex were HA-immobilized manner-dependent. At static state, the better thermal stability of interfacial H-bonding was granted to CD44 bound with the end-fixed but center-loosed HA rather than others (Figure 3e), and, under stretching, most of the force-enhanced interfacial H-bonds were presented to CD44 bound with the center-fixed HA chains rather than that of two fixed ends alone (Figure 5g–i and Figure 7e). The HA-binding residues consisting of Arg^155^, Arg^82^, Cys^81^, Tyr^46^, Ile^100^ and Asn^115^ were responsible for the mechanical stability of the complex (Figure 5g–i and Figure 7e), and all these residues except Asn^115^ were contributed to thermal stability of the complex (Figure 3e). Additionally, H-bonding of Tyr^166^ to Glu^52^ was required for maintaining conformational stability and preventing the complex from unfolding, as shown in the previous work [26,27].

It should be pointed out that there existed different binding modes of CD44 and HA, leading to diverse identified HA-binding residues in previous studies [41,42]. Instead of binding modes, we herein focused on the effects of HA-immobilized manner on CD44-HA interaction at static state or under tensile force. It was why a number of identified HA-binding residues was not shown in the present work. Moreover, the recombinant human CD44 protein (Met 1-Pro 220), 5 kDa HA chain and its immobilized manner in the present AFM experiment were not matched entirely with those in MD simulations, leading to a limitation on the present results. More single molecular experiments and MD simulations were expected for the immobilized manner-dependent interaction of CD44 with HA considering adhesion mode, HA weight, matrix stiffness and mechano-microenvironment. However, the present data from AFM measurements and SMD simulation did support each other partially but rationally and provide an argument for mechanical regulation mechanism and its molecular basis on CD44 binding with immobilized HA at static state or under tensile force.

In conclusion, we herein provided a novel MD-based computer strategy to investigate how the mechanical constraints on the HA chain regulate HA affinity to CD44, and found that, under mechanical constraints, the HA chain required a moderate flexibility in favor of thermal excitation-driven association rather than force-induced dissociation of CD44-HA complex. The end-fixed but center-loosed HA was in favor with CD44 at static state because of its moderate flexibility, which enabled the HA chain to match with CD44 under thermal excitation. Additionally, this end-fixed but center-loosed HA would become a disadvantageous convex one for preventing dissociation of the complex under stretching, while a shrunken HABD under tensile force enhanced strength and mechanical stability of the complex. These findings suggested a faster force-dependent reaction kinetics in adherence of CD44 to the HA matrix with moderate stiffness and loosed middle of the binding site also. Meanwhile, force-induced breakage of H-bonding of Tyr^166^ to Glu^52^ was a key event in force-enhanced adherence of CD44 to the HA matrix. This study argued a novel statement for the mechano-regulation mechanism and its molecular basis of CD44 binding to HA under diverse constraints and provided a novel insight into understanding the CD44-HA interaction-mediated cell inflammatory responses, tumor development and metastasis under mechanical microenvironments.

## 4. Materials and Methods

### 4.1. System Setup and Equilibrium in Molecular Dynamics (MD)

The crystal structure of the murine CD44 HABD (residue 24–173) in complex with HA sugar chain (residue 1175–1181) was taken from the Protein Data Bank (PDB) database (PDB code 2JCR). Two software packages, the NAMD 2.13 [43] for MD simulation and the Visual Molecular Dynamics (VMD) 1.9.2 [44] for visualization and modeling, were used. All missing hydrogen atoms and bonds (disulfide bonds, glycosidic bonds) were added by AUTOPSF, a plug-in of VMD. The complex was solvated in a rectangular (65.1 × 77.7 × 72.3 Å) TIP3P water box, into which 150 mM Na^+^ and Cl^−^ were added for achieving a system charge neutrality and mimicking the actual physiological environment. MD simulations were performed using periodical boundary condition and the CHARMM36 all-atom force field [45], along with cMAP correction for backbone, particle mesh Ewald (PME) algorithm for electrostatic interaction, a 12 Å cut off for electrostatic and van der Waals interaction, at a timestep of 2 fs. The system consisted of 33,931 atoms and was subjected to an energy minimization for three consecutive durations of 15,000 timesteps, firstly with protein and sugar chain being fixed, secondly with only the heavy atoms of the protein being fixed and finally with all atoms free, at 0 K. After energy minimization, the system was heated gradually from 0 to 310 K in 0.1 ns and then equilibrated three times for 40 ns with pressure and temperature control. The temperature was maintained at 310 K using Langevin dynamics, and the pressure was maintained at 1 atom by the Langevin piston method. The best stable structure in equilibrium was chosen as the initial conformations for subsequent free molecular dynamics (FMD) simulations and steered molecular dynamics (SMD) simulations.

### 4.2. Free and “Ramp-Clamp” MD Simulation

To model the diverse attachments of HA to the extracellular matrix, four different systems, named S0, S1, S2 and S3, were used in FMD simulation. FMD simulations were performed in a microcanonical ensemble, which means that the temperature and pressure were not controlled at a preset level as in the equilibrium to better display the conformation changes and thermal stability of the protein [46]. S0 was same as the equilibrated system without any mechanical constraint on HA. Using the equilibrated system, S1 was set up just through fixing the C2 atom of HA GlcUA1178, and S2 did so by fixing C2 atoms, one on GlcNAc1175 and the other on GlcNAc1181, while S3 was built up through fixing three C2 atoms on GlcNAc1175, GlcUA1178 and GlcNAc1181, respectively. All runs were performed thrice on each system for 100 ns. Additionally, the so-called “ramp-clamp” SMD simulations, a force-clamp MD simulation followed a force-ramp one, were performed on systems S1, S2 and S3 to model regulation of mechanical constraints on HA on force-induced unbinding and conformation changing of HA-ligated CD44. The C_α_ atom of the CD44 HABD C-terminal residue Ile^173^ was steered along pulling direction from C2 atom of GlcUA1178 to the pulled atom for each system with mechanical constraint on HA. The virtual spring, connecting the dummy atom and the steered atom, had a spring constant of 13.90 pN/Å. The complex was pulled over 30ns thrice with time step of 2 fs and a constant velocity of 5 Å/ns. Once tensile force arrived at 25 pN, the SMD simulation was transformed from the force-ramp mode to a force-clamp one, at which time the complex was stretched with the given constant tensile force for the following 40 ns. Each event of hydrogen bonding under stretching was recorded to examine the involved residues and their functions. All the atomic coordinates were recorded, and visualization of trajectories were presented using VMD.

### 4.3. Data Analysis in MD

All data from MD simulation were analyzed with VMD tools. The C_α_-root-mean-square deviation (RMSD) of the complex, the C-root-mean-square fluctuation (RMSF) of the bound HA, and the solvent-accessible surface area (SASA) (with a 1.4 Å probe radius) of the binding site were measured for each run to mimic the structural stability, flexibility and change, as well as the interfacial geometry characteristics of the binding site. The number of hydrogen bonds (H-bonds) on the complex interface was read with cutoff donor–acceptor distance of 3.5 Å and cutoff donor-hydrogen–acceptor angle of 30°, respectively, while forming and maintaining of a salt bridge had a requirement of the distance (<4 Å) between any one of the oxygen atoms in acidic residues (Asp or Glu) and the nitrogen atoms in basic residues (Lys or Arg). An occupancy (or survival ratio) of a hydrogen bond or a salt bridge was measured by the fraction of bond survival time in the simulation period. ***P***_D_, the probability of ligand dissociation from receptor, was used herein to approximately evaluate affinity of the receptor to its ligand, like our previous works [47]. The dissociation probability ***P***_D_ is calculated by survival ratios of hydrogen bonds across the binding site, with the assumption that each hydrogen bonding event on the binding site is independent and fully responsible for the integrin affinity to talin.

### 4.4. Molecules and Reagents in Atomic Force Microscopy (AFM)

Recombinant human CD44 protein (Met 1-Pro 220) was obtained from Sino Biological Inc. (Beijing, China). An amount of 5 kDa unlabeled HA came from Lifecore (Chaska, MN, USA), while 5 kDa HA with one or two biotins were customized at Creative PEGWorks (Chapel Hill, NC, USA). Additionally, streptavidin was from Thermo Fisher Scientific^TM^ (Waltham, MA, USA). All the other reagents were of analytical grade or the best grade available.

### 4.5. Measurement of Adhesion Frequency, Rupture Force and Lifetime

To examine CD44-HA interaction, the AFM cantilever tip (MLCT; Bruker AFM Probes) was incubated with CD44 (15 μL, 10 μg/mL) or 2% bovine serum albumin (BSA) (Control) overnight at 4 °C. After rinsing with phosphate-buffered saline (PBS), the tip was incubated with PBS containing 2% BSA for 1 h at room temperature to block nonspecific binding. The Petri dish was functionalized in two different manners, like the treatment with physical adsorption and biotin–avidin interaction [29]. In the first manner, HA (15 μL, 100 μg/mL) without biotin tag was absorbed to a small spot on a Petri dish overnight at 4 °C, then the functionalized dish was washed 2 times with PBS. In the second manner, a sufficient amount of streptavidin was absorbed on a Petri dish first, and then the streptavidin-treated disk was incubated overnight at 4 °C and washed 2 times with PBS and incubated again with PBS containing 2% BSA for 1 h at room temperature; subsequently, HA (15 μL, 100 μg/mL) with biotin tag was absorbed to a small spot on the streptavidin-treated disk for one hour at room temperature, and this dish was filled with PBS containing 2% BSA and rested at room temperature for 1 h before measuring.

During each adhesion frequency and rupture force measurement cycle [48], a Petri dish was driven by a piezoelectric translator (PZT) into contact with a cantilever tip, which retracted slightly immediately after reaching the set point (0.5 V) and held there for 500 ms to facilitate the formation of the bond formation, after which it retracted directly to the initial position at a speed of 100 nm/s along the z direction. An event that the voltage signal significantly increased during retraction was considered as an adhesive event. A total of 100 cycles were repeated at each spot, and the adhesion frequency was calculated; the rupture force of each adhesion events was also counted. More than 3 different spots were detected for each sample. For lifetime measurement, retraction would stop when a preset level was reached, clamping the force at that level until the tensile force broke and further retracted to its initial position. The bond lifetimes at desired forces were measured from the force–time curves.

### 4.6. Measurement of Elastic Modulus

The elastic modulus ***E*** of HA on the Petri dish was examined via AFM measurements, in which HA was coated on the Petri dish with or without streptavidin-treatment, as described above, and the nonspecific binding was blocked via incubating AFM tip with BSA for 1h at room temperature in the absence of CD44. Based on the Hertz model, the elastic modulus *E* of HA were extracted from fitting results of the force–displacement curves from AFM measurements [49]. For the Hertz model, the relationship between load and indentation depth was:(1)$$F=\frac{2}{\pi }\tan\alpha \frac{E}{1-v^{2}}\delta^{2}$$
where *F* was the force on the conical AFM tip, *α* the half-opening angle of the tip, *E* the elastic modulus of the measured material, and *δ* the indentation depth, which could be defined as the distance travelled by the piezoelectric ceramic. Poisson’s constant ***v*** was taken as 0.5 for the biological materials.

The flow diagram for studying mechanical constraints on CD44-HA interaction via MD simulation and AFM experiments was provided (see the Appendix A).

### 4.7. Statistical Analysis

Data are presented as mean values ± the standard error of the mean (SEM). Statistical significance was analyzed by one-way (or two way) analysis of variance (ANOVA) followed by Tukey’s multiple comparisons test; * means *p* < 0.05, ** means *p* < 0.01, *** mean *p* < 0.001, **** *p* < 0.0001, ns means not significant

## Figures and Tables

**Figure 1 ijms-24-02243-f001:**
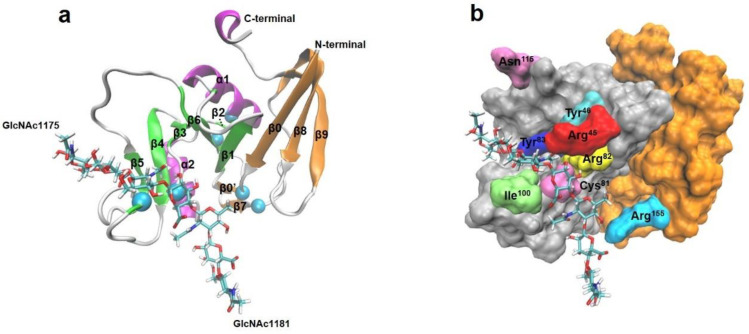
Structure of the CD44(HABD)-HA complex. (**a**) The structure of complex of HA with the HA-binding domain (HABD) of CD44 (PDB code 2JCR) in Newcartoon representation. The HABD consists of Link and extension modulus, the Link module consists of β1 α1 β2 α2 β3 β4 β5 β6, while the extensive one includes β0 β’0 and β7 β8 β9 as well C-terminal. The β-sheets are colored with green for the Link modulus and orange for the extensive modulus. The α-helices and the loop between two secondary structures were painted with magenta and white, respectively. HA bound to HABD with two GlcUAc ends, GlcUAc1175 and GlcUAc1181, was shown in licorice representation. (**b**) The complex binding site in surface representation, where the CD44 Link module (silver), N- and C-terminal Link extension (orange) and the critical amino acids (Arg^45^, Tyr^46^, Cys^81^, Arg^82^, Tyr^83^, Ile^100^, Asn^115^ and Arg^155^) on the binding site of CD44 were represented, shown in the representation.

**Figure 2 ijms-24-02243-f002:**
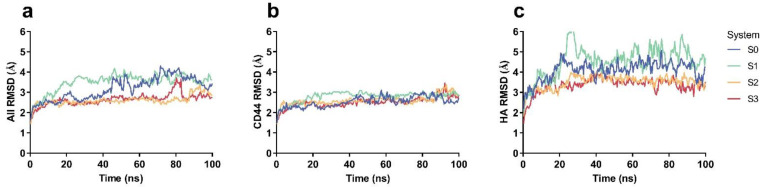
The mean time course of RMSD for CD44-HA under different mechanical constraints. The data of RMSD for (**a**) the CD44-HA complex, (**b**) the bound CD44 alone and (**c**) the bound HA alone were from three independent runs of 100 ns. Four systems, such as system S0 (blue), S1 (green), S2 (yellow) and S3 (red), were set up by fixing zero, one, two and three C2 atoms on HA in the equilibrated system, respectively. The fixed C2 atom in S1 was on GlcUA1178, the two fixed C2 atoms in S2 were located, respectively, at GlcNAc1175 and GlcNAc1181, while the three C2 atoms in S3 were in GlcNAc1175, GlcUA1178 and GlcNAc1181, respectively.

**Figure 3 ijms-24-02243-f003:**
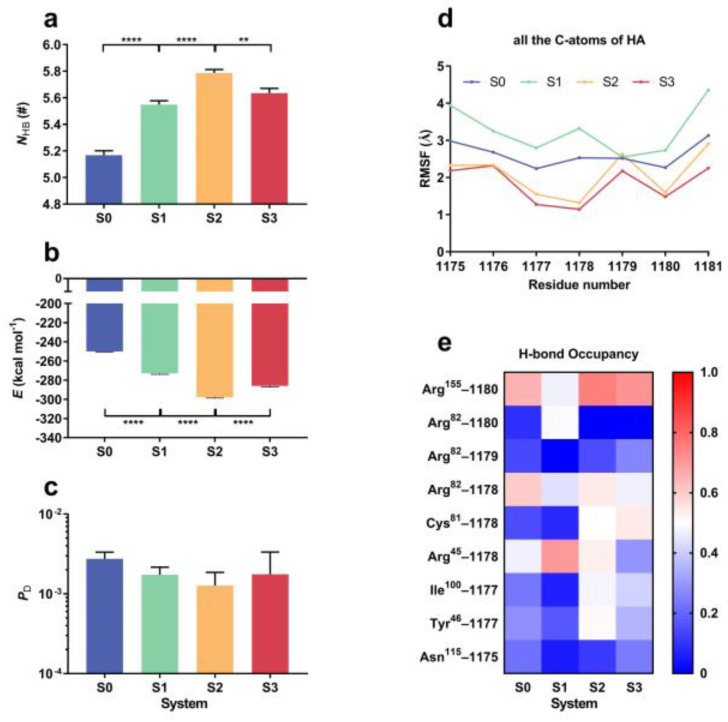
Regulation of HA constraint manner on HA-CD44 interaction. Plots of (**a**) the number of the interfacial H-bond (***N***_HB_), (**b**) the interfacial interaction energy (***E***) and (**c**) the complex dissociation possibility (***P***_D_) against HA constraint manner in system S0, S1, S2 or S3. (**d**) The RMSF of residues of HA on different constraints in system S0, S1, S2 and S3. The RMSF was generated for all six carbon atoms of each monosaccharide of the HA. (**e**) the heatmap of interfacial H-bond occupancies and their involved residue pairs. Each date was the mean from three independent runs over 100 ns for each given HA constraint manner. All data are shown as means ± SEM. Statistical significance was analyzed by one-way ANOVA followed by Tukey’s multiple comparisons test, **** *p* < 0.0001, ** *p* < 0.01, or ns., not significant.

**Figure 4 ijms-24-02243-f004:**
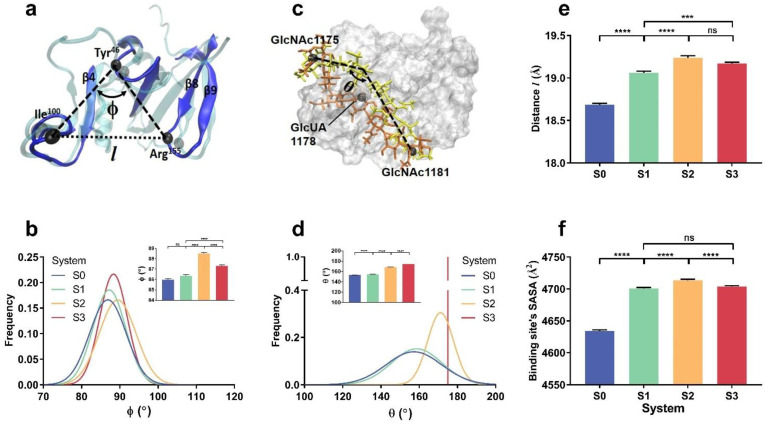
Constraint-mediated conformational change of CD44-HA during FMD simulations. (**a**) The bag-like CD44 binding site. Of the triangle with three vertices (the C_α_-atoms of Arg^155^ and Ile^100^ and Tyr^46^) on CD44 HABD, the distance (*l*) between two C_α_-atoms of Arg^155^ and Ile^100^ and the cross angle (ϕ) of two link lines, one from Arg^155^ to Tyr^46^ C_α_-atom and another from Ile^100^ to Tyr^46^, are used to quantify HABD deformation under different mechanical constraints on HA. (**b**) Distribution of the cross angle ϕ from three runs of 100 ns under three different constraints in systems S0, S1, S2 and S3. (**c**) The HA bend angle θ (the cross angle of two link lines, one from GlcUA1178- to GlcNAc1175- C2 atom and another from GlcUA1178- to GlcNAc1181- C2 atom) and (**d**) its distribution from three runs of 100 ns in systems S0, S1, S2 and S3. Plots of (**e**) distance (*l*) between two C_α_-atoms of Arg^155^ and Ile^100^ and (**f**) the SASA of HABD binding site against the HA constraint manner in systems S0, S1, S2 or S3. All data shown are means ± SEM. Statistical significance was analyzed by one-way ANOVA followed by Tukey’s multiple comparisons test, **** *p* < 0.0001, *** *p* < 0.001, or ns., not significant.

**Figure 5 ijms-24-02243-f005:**
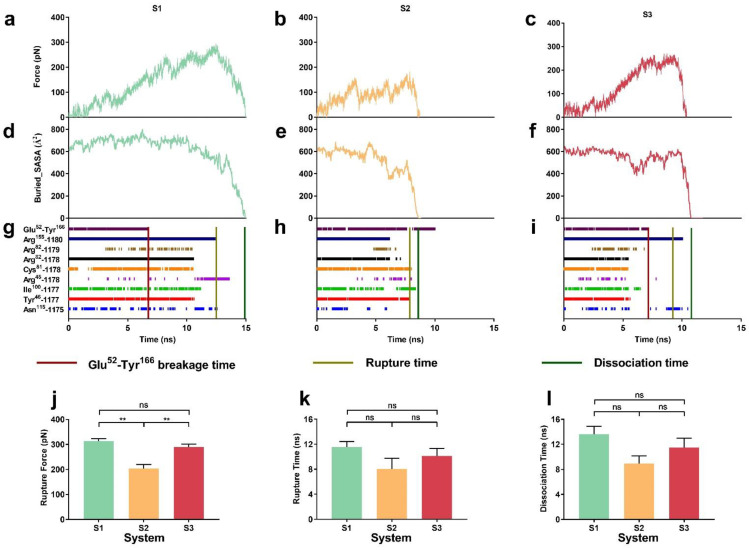
The time courses of tensile force, buried SASA and interfacial H-bonds along the pull-induced dissociation pathway of CD44-HA complex in the systems S1, S2 and S3 at pulling velocity of 5 Å/ns. The representative force–time curves (**a**–**c**), the buried SASA-time curves (**d**–**f**) and the survival patterns of interfacial H-bonds (**g**–**i**) of the pulled complex in systems S1, S2 and S3, respectively. The rupture force (**j**), rupture and dissociation time (**k**,**l**) of the complex in the systems S1 and S2 as well as S3, respectively. All data shown came from three dependent SMD runs of 15 ns at pulling velocity of 5 Å/ns and were means ± SEM. Statistical significance was analyzed by one-way ANOVA followed by Tukey’s multiple comparisons test, ** *p* < 0.01, n.s., not significant.

**Figure 6 ijms-24-02243-f006:**
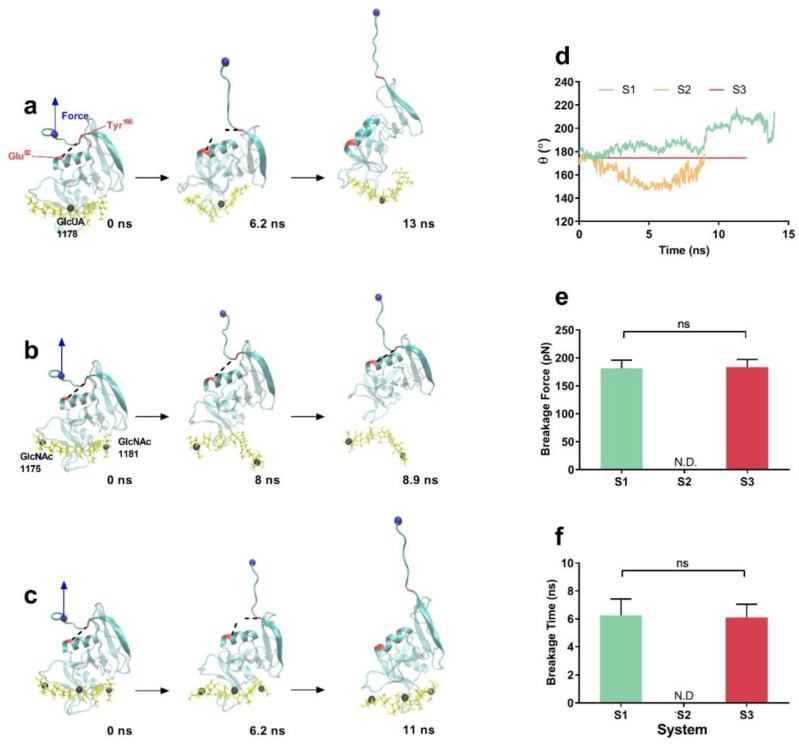
Pull-induced allostery of the constrained complex at pulling velocity of 5 Å/ns. (**a**–**c**) The typical snapshots of the pulled CD44 (cyan) bound to HA (yellow) for molecular systems S1, S2 and S3 at different pull times from of 0 to 15 ns. The fixed atoms were colored with black, and the blue sphere denoted the steered atom, while the black dot lines indicated the H-bond linker between Glu^52^ and Tyr^166^. (**d**) The time course of the HA bend angle θ in systems S1, S2 and S3. (**e**,**f**) The breakage force and time plots of the H-bond between Glu^52^ and Tyr^166^ for three systems. N.D. meant no data of either breakage force or time and exhibited a stable H-bonding of Glu^52^ to Tyr^166^ in system S2 during the pull-induced dissociation. All data shown are means ± SEM. Statistical significance was analyzed by one-way ANOVA followed by Tukey’s multiple comparisons test, ns, not significant.

**Figure 7 ijms-24-02243-f007:**
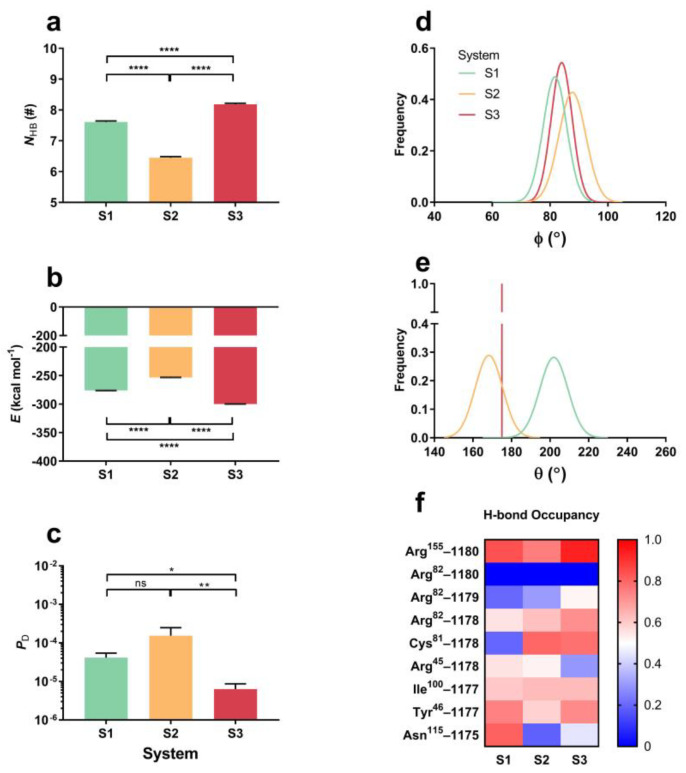
The force induced strong engagement between CD44 and HA, and the complexes had different mechanical stability at different HA flexibilities. Plots of (**a**) the number of the interfacial H-bond (***N***_HB_), (**b**) the interfacial interaction energy (***E***) and (**c**) the complex dissociation possibility (***P***_D_) against HA constraint manner in system S1, S2 or S3 under a constant force of 25 pN for three runs at 40 ns “force-clamp” SMD simulations. (**d**,**e**) Distribution of the cross angle ϕ and the HA bend angle θ from three runs of 40 ns under three different constraints in systems S0, S1, S2 and S3. (**f**) Variation of the occupancy of H-bonds with the constraints. The color bars marked the values of occupancy. All data shown are means ± SEM. Statistical significance was analyzed by one-way ANOVA followed by Tukey’s multiple comparisons test, **** *p* < 0.0001, ** *p* < 0.01, * *p* < 0.05; ns, not significant.

**Figure 8 ijms-24-02243-f008:**
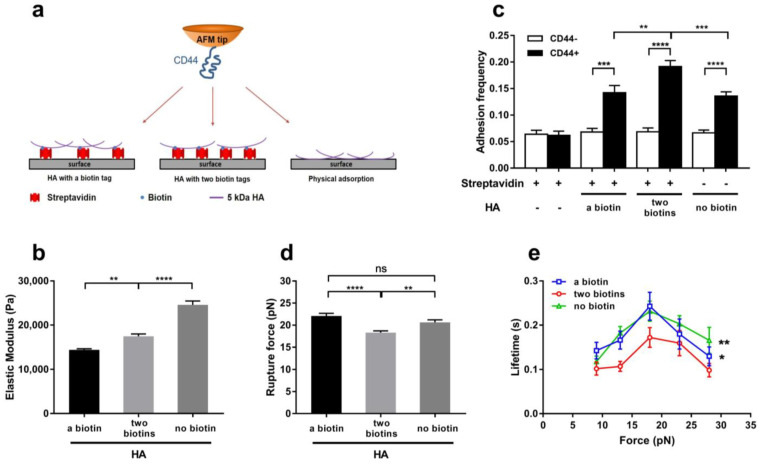
Variation of adhesion frequency, rupture force and lifetimes of CD44 on HA-coated substrate versus the HA-immobilized manner. (**a**) Schematic for single molecular AFM experiments. An amount of 100 μg/mL 5 kDa HA on substrate was immobilized through one and two biotin–avidin linkage or physical absorption, respectively. (**b**) Elastic modulus of HA-coated substrates with three different HA-immobilized treatments. (**c**) Specificity and frequency of adhesion of the recombinant protein CD44-functionalized tips to the immobilized HA on substrates. “CD44+” and “CD44−” denoted the CD44 and BSA coated cantilever tips, respectively. (**d**) Rupture forces and (**e**) lifetime of CD44 on HA-coated substrates using different HA immobilized treatments. All data shown are means ± SEM. Statistical significance was analyzed by one-way (or two-way) analysis of variance (ANOVA) followed by Tukey’s multiple comparisons test, **** *p* < 0.0001, *** *p* < 0.001, ** *p* < 0.01, * *p* < 0.05, ns means not significant.

## Data Availability

The data that support the findings of this study are available from the corresponding author upon reasonable request.

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
