# Peer review of "Moderate Constraint Facilitates Association and Force-Dependent Dissociation of HA-CD44 Complex"

_ijms, 2023, doi:10.3390/ijms24032243_

Round 1

Reviewer 1 Report

Yao et al., studied the interaction between CD44 and HA using MD simulations as well as atomic force microscope (AFM) measurements using four different constraint systems. The study is well-designed, and the authors explained their findings in a well-defined way. Therefore, can be considered for publication after minor changes. A few suggestions for the improvement of the study are:

In Figure 1 please show the 3D-intermolecular interactions between both molecules.

Instead of a PDB code, a PDB id should be used.

Why was the equilibration performed for 40 ns and how was this time decided?  Since the equilibration plot displayed similar fluctuations.

Why rectangular box was considered for MDS please explain.

On what basis four systems of HA were prepared for the MDS study? Please explain

Can authors explain Free molecule dynamics to readers with a short note?

Why the 3D-intermolecular interactions were not provided after 100 ns MDS for all the systems?

Please improve the typing and grammatical errors throughout the Manuscript. For example, in Section 4.1-line number 528, something missing here, Section 4.3 line 571- squared to square, rewrite section 4.4.

Reviewer 2 Report

The authors focused on the interaction of CD44 and HA using molecular dynamics simulations and atomic formic microscopy. By modulating various of HA chains flexibility, a lot of different properties and interactions were characterized, including thermal stability, mechanical stability, etc. The regulation mechanism and molecular basis were illustrated and hence new insights were potentially provided. However, there are still several issues:

1. The organization of this paper needs to be modified. The conclusion part should be added or separated. In either case, the highlight of this work should be presented. 

2. What is the novelty and significance of this work? I didn't see much clear explanation in the introduction part. A short description needs to be supplied. 

3. The second and third paragraph of the introduction part is somewhat unnecessary, as it's not the authors' job to make some introductory comments. I recommend the authors to combine these two parts and make it more concise and clear. 

Reviewer 3 Report

The manuscript entitled “Moderate Constraint Facilitates Association and Force-dependent Dissociation of HA-CD44 Complex” by Yao and co-authors performed the MD simulation of CD44 in complex with HA using NAMD. Authors have explored the interactions of CD44 HABD to HA. Authors have performed atomic force microscope (AFM) experiments at single molecular level, free and steered molecular dynamics (MD) simulations with structure of CD44 HABD-HA complex to unravel the mechanical regulation mechanism and molecular basis for CD44 interaction under various mechanical microenvironment. All the figures are relevant and self explanatory. I have a few queries/suggestions that need to be addressed prior to acceptance of the manuscript. I recommend minor revision of the manuscript. Here are my queries for the manuscript:-

1.    Manuscript need to be screen for typos and grammatical.

2.    Author should incorporate a graphical representation or flow chart to represent the overall workdone and outcome of the study that will be beneficial for the future readers.

3.    In Figure2, it is very hard to see the color differences of S0, S1, S2, and S3, as this figure has three sub figure in a single row. Authors can represent these sub panels big by arrangement of these figure in a same column instead of row.

4.    From Figure 2a and 2c, it is clearly seen that HA has higher flexibility in S0 and S1 systems than S2 and S3. What is the reason of this dynamic movement of HA ?

5.    In Figure 3d, authors have shown the RMSF for amino acid residues of 1175 to 1181. From the graph, it seems that there is no significant mobility difference in this region in all the systesm. Authors should show the RMSF for full protein also to visualize the flexibility in complex. Authors need to mention that RMSF was generated for C alpha atoms or backbone atoms. 

Round 2

Reviewer 2 Report

The quality of this paper has been improved.